# Effect of Carbamylated Erythropoietin in a Chronic Model of TNBS-Induced Colitis

**DOI:** 10.3390/biomedicines11092497

**Published:** 2023-09-09

**Authors:** Inês Silva, Mário Gomes, Carolina Alípio, Jéssica Vitoriano, João Estarreja, Priscila Mendes, Rui Pinto, Vanessa Mateus

**Affiliations:** 1H&TRC—Health & Technology Research Center, ESTeSL—Escola Superior de Tecnologia da Saúde, Instituto Politécnico de Lisboa, 1990-096 Lisbon, Portugal; ines.silva@estesl.ipl.pt (I.S.); mjgomes@estesl.ipl.pt (M.G.); carolinaralipio@gmail.com (C.A.); estarreja.20112014@gmail.com (J.E.); priscila.mendes@estesl.ipl.pt (P.M.); 2iMed.ULisboa, Faculdade de Farmácia, Universidade de Lisboa, 1349-017 Lisbon, Portugal; rapinto@ff.ulisboa.pt; 3Centro de Química Estrutural, Institute of Molecular Sciences, Instituto Superior Técnico, Universidade de Lisboa, 1349-017 Lisbon, Portugal; 4Joaquim Chaves Saúde, Laboratório de Análises Clínicas, Miraflores, 1495-069 Algés, Portugal

**Keywords:** inflammatory bowel disease, TNBS-induced colitis, carbamylated erythropoietin, rodents

## Abstract

Background: Inflammatory bowel disease (IBD) is a public health issue with a growing prevalence, which can be divided into two phenotypes, namely Crohn’s disease (CD) and ulcerative colitis (UC). Currently, used therapy is based only on symptomatic and/or palliative pharmacological approaches. These treatments seek to induce and maintain remission of the disease and ameliorate its secondary effects; however, they do not modify or reverse the underlying pathogenic mechanism. Therefore, it is essential to investigate new potential treatments. Carbamylated erythropoietin (cEPO) results from the modification of the Erythropoietin (EPO) molecule, reducing cardiovascular-related side effects from the natural erythropoiesis stimulation. cEPO has been studied throughout several animal models, which demonstrated an anti-inflammatory effect by decreasing the production of several pro-inflammatory cytokines. Aim: This study aimed to evaluate the efficacy and safety of cEPO in a chronic TNBS-induced colitis model in rodents. Methods: Experimental colitis was induced by weekly intrarectal (IR) administrations of 1% TNBS for 5 weeks in female *CD-1 mice*. Then, the *mice* were treated with 500 IU/kg/day or 1000 IU/kg/day of cEPO through intraperitoneal injections for 14 days. Results: cEPO significantly reduced the concentration of alkaline phosphatase (ALP), fecal hemoglobin, tumor necrosis factor (TNF)-α, and interleukin (IL)-10. Also, it demonstrated a beneficial influence on the extra-intestinal manifestations, with the absence of significant side effects of its use. Conclusion: Considering the positive results from cEPO in this experiment, it may arise as a new possible pharmacological approach for the future management of IBD.

## 1. Introduction

Inflammatory bowel disease (IBD), including Crohn’s disease (CD) and Ulcerative colitis (UC), are chronic idiopathic conditions that cause inflammation of the gastrointestinal tract. In the last decade, IBD has emerged as a global disease, becoming a public health problem all around the world [1]. These disorders affect over 2 million individuals in North America, 3.2 million in Europe, and millions more worldwide [2]. In recent years, there has been increasing interest in the cEPO molecule due to its protective and anti-inflammatory properties across various diseases [3]. cEPO is derived from the modification of the EPO molecule and is a non-erythropoietic erythropoietin carbamoyl derivative [3]. EPO, a glycoprotein hormone primarily produced in the fetal liver and adult kidney under hypoxic conditions, plays a crucial role in regulating the production, survival, and differentiation of red blood cells into mature erythrocytes within the human and murine bone marrow through its binding to the EPO receptor (EPOR) [4,5,6,7,8,9]. Despite its benefits, EPO can also lead to high side effects due to its stimulation of erythropoiesis, posing a risk of severe cardiovascular events [4,10]. To mitigate these potential risks and side effects, the utilization of the cEPO molecule has been proposed as an alternative approach [11]. By modifying the EPO molecule, it is possible to maintain similar pharmacokinetic characteristics while avoiding binding to the conventional homodimeric EPOR, thus preventing erythropoietic side effects [3]. Instead, cEPO exerts its effects by binding to a heterodimeric EPO common-β receptor (CD131), which is also shared by GM-CSF, IL-3, and IL-5 without activating JAK-2 [12]. The activation of the heteroreceptor, composed of EPOR and CD131, leads to protective responses characterized by anti-inflammatory and anticoagulant effects similar to those observed with EPO [13,14]. Both EPO and cEPO have been shown to decrease the production of inflammatory mediators like TNF-α, IL-1β, and lymphocytes, which are associated with conditions such as IBD, and increase the levels of IL-10, indicating that cEPO can effectively counteract inflammation [15]. As a result, cEPO demonstrates beneficial effects on cell proliferation, the inflammatory process, angiogenesis, antiapoptotic outcomes (by inhibiting caspase-3), and neurogenesis, all while avoiding the side effects related to erythropoiesis [10,16,17,18].

Consequently, cEPO can be a possible new strategy for IBD, maintaining EPO efficacy and improving its safety [4]. Therefore, in order to study the efficacy and safety of cEPO in IBD, it will be administrated in a chronic TNBS-induced mouse model of colitis.

## 2. Materials and Methods

### 2.1. Material

2,4,6-Trinitrobenzene sulfonic acid (TNBS 5%) was purchased from Sigma–Aldrich Chemical, Alges, Portugal, EUA; Ketamine (Ketamidor^®^ 100 mg/mL) and Xylazine (Sedoxylan^®^ 20 mg/mL) were obtained from Bio2, Quinta do Anjo, Portugal. Erythropoietin (Eprex^®^ 2000 IU/0.5 mL) was purchased from Janssen-Cilag, Porto Salvo, Portugal. Boric acid (H_3_BO_3_), hydrochloric acid (HCl), potassium cyanate (KOCN), phosphate-buffered saline (PBS), sodium chloride (NaCl), sodium borate (Na₂[B₄O₅(OH)₄]·8H₂O) and sodium bicarbonate (NaHCO₃) were obtained from Sigma Chemical Co., EUA. Elisa assay kits for cytokines measurements were obtained from Hycult Biotechnology, Uden, The Netherlands, EUA^®^. ADVIA^®^ kit for biochemical markers was purchased from Siemens Healthcare Diagnostics, Amadora, Portugal^®^.

### 2.2. Animals

Throughout this study, female *CD-1 mice* aged 6 weeks and weighing 30 g were employed. The animals underwent a one-week acclimatization period and were housed in standard polypropylene cages with unrestricted access to food and water in the animal facility. The environmental conditions, including temperature, humidity, and lighting, were carefully controlled during the experiment. Specifically, the *mice* were maintained under a controlled 12 h light/dark cycle at a temperature range of 18–23 °C and humidity between 40 and 60%. All aspects of animal care and experimental procedures adhered strictly to the internationally recognized principles for the use and welfare of laboratory animals, as stated in Directive 2010/63/EU [19], which has been implemented in Portuguese legislation by Directive Law 113/2013. The experiment received ethical approval from the Animal Welfare Body (Órgão Responsável pelo Bem-Estar Animal—ORBEA) at the Faculty of Pharmacy, University of Lisbon, with the assigned code number 3/2020. Additionally, it obtained further approval from the Directorate-General for Food and Veterinary (Direção Geral de Alimentação e Veterinária—DGAV) on 6 November 2020.

### 2.3. Trinitrobenzene Sulfonic Acid-Induced Colitis

The induction of chronic colitis was developed through weekly intracolonic administrations of TNBS 1% by enema. The animals were left unfed 24 h before each administration day to facilitate the intracolonic administrations, including control groups, to promote equal conditions between all groups. On the induction days, namely days 0, 7, 14, 21, and 28, *mice* were anesthetized with an intraperitoneal injection of 40 µL of a solution containing injection of Ketamine 100 mg/Kg + Xylazine 10 mg/Kg. Then, 100 µL of TNBS 1%, the induction agent, dissolved in 50% ethanol, was instilled by intracolonic administration, with the help of a cannula, carefully inserted until 4 cm in the colon. Then, *mice* were kept for 1 min in a Trendelenburg position to avoid rectal reflux of TNBS.

### 2.4. Carbamylated Erythropoietin Synthesis

To initiate the synthesis of cEPO, 100 mL of borate buffer with a pH of 8.9 was combined with 1 mL of EPREX^®^. The pH was then adjusted using HCl. Next, 2 M of KOCN was introduced into the mixture, and the solution was left to incubate at 37 °C for 20 h. After the incubation period, the solution underwent filtration in amicon tubes (4 mL) for 20 min. Subsequently, 2 mL of PBS was added and filtered again using amicon tubes for 15 min. Following this, 2 mL of 0.9% NaCl solution was added to the mixture and filtered in amicon tubes. This process was repeated once more. Finally, an additional 2 mL of NaCl was added to the solution to achieve a concentration of 2000 IU/0.5 mL.

The synthesis and formulation of cEPO involved several steps. Initially, sodium borate was added and incubated with the EPREX^®^ 2000 IU/0.5 mL molecule to act as a buffer solution with a pH of 8.9. Subsequently, potassium cyanate was introduced into the solution, which was then left to incubate for 20 h at 37 °C. After the incubation period, the solution underwent filtration using amicon tubes. Additional steps included adding PBS and repeating the filtration process. Finally, NaCl was added to the solution and filtered once more, resulting in a cEPO solution at a concentration of 2000 IU/0.5 mL. The process of carbamoylation occurs through the covalent binding of potassium cyanate to proteins [20].

### 2.5. Determination of Carbamylation Rate

A solution of cEPO at a concentration of 500 IU/mL was prepared. To this solution, 1 mL of 4% NaHCO₃ with a pH of 8.4 was added. Subsequently, 50 µL of TNBS at 0.1% concentration was introduced. The mixture was then incubated at 37 °C for 1 h. The absorbance of this solution was measured at 335 nm using the Pierce method. The same procedure was repeated for a sample of EPO at a concentration of 500 IU/mL. As a reference, a solution containing 1 mL of NaCl, 1 mL of NaHCO₃, and 50 µL of TNBS at 0.1% concentration was used. The absorbance of the cEPO and EPO solutions was recorded. Additionally, 50 µL of 1% TNBS, 1 mL of NaCl, and 1 mL of 4% NaHCO₃ were added to both the cEPO and EPO solutions. The NaHCO₃ served as a buffer solution with a pH of 8.4, facilitating the deprotonation of lysine residues. After the buffer solution was added, TNBS was introduced, and the mixture was incubated at 37 °C for one hour before measuring the absorbance.

### 2.6. Experimental Groups

The *mice* were categorized into six groups based on the primary objective of this study. The experimental groups included the TNBS group (*n* = 5)—Disease control group, in which *mice* received five weekly intrarectal (IR) administrations of 100 µL of TNBS at 1%; the TNBS + cEPO 500 group (*n* = 20) and the TNBS + cEPO 1000 group (*n* = 20)—treated groups, where *mice* were induced with colitis through the administration of TNBS over five weeks and treated daily with 500 IU/kg and 1000 IU/kg of cEPO by IP administration for 14 days, respectively; cEPO 1000 group (*n* = 10)—drug control group, where *mice* were subjected to a daily IP administration of cEPO 1000 IU/kg, for 14 days; Ethanol group (*n* = 5)—TNBS’s vehicle control group, where *mice* were subjected to five weekly IR administrations of 100 µL of Ethanol 50%; and Sham group (*n* = 5)—control group, where *mice* were subjected to five weekly IR administrations of 100 µL of NaCl 0.9%.

### 2.7. Monitoring of Clinical Signs

The animals were observed daily, and their body weight, stool consistency, morbidity, and mortality were monitored throughout this study.

### 2.8. Macroscopic Evaluation

The macroscopic evaluation of colitis, including the assessment of hyperemia, ulcers, inflammation, the number of sites with ulceration and/or inflammation, and its extent, was conducted following the methodology described by Morris et al. in 1989 [21].

### 2.9. Biochemical Markers and Cytokines

Serum from the collected blood samples was separated by centrifugation at 3600× *g* for 15 min. Analysis of serum was conducted to evaluate several parameters, such as ALP and extra-intestinal manifestations, urea, creatinine, and alanine aminotransferase (ALT). All of them are evaluated spectrophotometrically.

The pro-inflammatory cytokine TNF-α and the anti-inflammatory cytokine IL-10 were measured and expressed as pg/mL.

Additional analyses were also conducted, such as fecal hemoglobin, which is evaluated using a quantitative method by immunoturbidimetry as an index of hemorrhagic focus [22].

### 2.10. Histopathological Analysis

The colon samples were preserved in 10% phosphate-buffered formalin, subjected to standard processing for paraffin embedding, and sectioned at 5 µm thickness. Hematoxylin and eosin staining were performed to visualize general tissue morphology. Additionally, Masson’s trichrome staining was utilized to enhance the likelihood of detecting fibrosis. Evaluation of the sections was focused on the distal colon based on the adapted criteria of Seamons and colleagues (2013) [23]. The histopathological score of lesions was partially scored (0–4 increasing in severity) with some parameters, namely: (1) the presence of tissue loss/necrosis; (2) the severity of the mucosal epithelial lesion; (3) inflammation; (4) extent 1—the percentage of the intestine affected in any manner; and (5) extent 2—the percentage of intestine affected by the most severe lesion. The colitis severity was calculated by summing the individual lesions and the extent scores, promoting a final colitis score (max score = 20). This task was carried out by two independent histopathologists from the Institute of Gulbenkian Science (IGC), blinded to the treatment groups.

### 2.11. Statistical Analysis

The findings were presented as the mean ± SEM of N observations, with N representing the number of animals studied. Data analysis was carried out using GraphPad 5.0 (GraphPad, USA) software. To determine the statistical significance between the TNBS and control groups, a one-way ANOVA was employed, followed by Tukey’s post hoc test for multiple comparisons. A *p*-value below 0.05 was considered statistically significant.

## 3. Results

### 3.1. Carbamylation of Erythropoietin

EPO (50 mU) was carbamylated as described by Horkko et al. (1992) with slight modifications [24]. In total, 0.5 mL of a Borate Buffer Solution pH 8.9 was added to 1 mL of EPO (4000 IU/mL). Potassium cyanate was added to obtain a solution with 2.0 M. This solution was incubated at 37 °C for 20 h. These procedures were followed by filtration with Amicon^®^ filters. In total, 2 × 2 mL of phosphate-buffered saline (PBS), pH 7.4, was added, and the solution was filtered, each time, with Amicon^®^. Finally, 2 mL of NaCl 0.9% was added, and the solution was filtered.

The extent of carbamylation was monitored by following the loss of free amino groups using trinitrobenzene sulfonic acid. To 1 mL of EPO (500 units/mL in normal saline) with 1 mL of 4% sodium hydrogen carbonate, pH 8.4, 50 mL of 0.1% trinitrobenzene sulfonic acid was added and the solution was incubated for 1 h at 37 °C. Absorbance was then measured at 340 nm against a sample blank, and the trinitrobenzene sulfonic acid reactivity was expressed as a percentage of the absorbance obtained for the non-carbamylated EPO. The determination of the carbamylation rate resulted in a percentage of 98.9%.

### 3.2. Monitoring of Clinical Signs

The *mice* were subjected to daily monitoring throughout the experimental procedure. Various parameters, including morbidity, stool consistency, anus appearance, and weight, were assessed. In the TNBS group, the *mice* displayed alterations in intestinal motility, characterized by diarrhea and soft stools, accompanied by mild edema of the anus. Similar clinical signs were observed in both cEPO treatment groups but with reduced severity. Notably, the cEPO1000, ethanol, and sham groups did not exhibit any discernible changes.

### 3.3. Fecal Hemoglobin

Fecal hemoglobin was measured and compared between all the experimental groups (Figure 1). This biomarker allows the evaluation of the intensity of hemorrhagic focus and the influence of cEPO treatment. The TNBS group presented a significantly higher concentration of fecal hemoglobin in comparison with control groups cEPO1000, Ethanol, and Sham (6.60 ± 0.76 μmol/g feces vs. 2.14 ± 0.07 μmol/g feces vs. 1.70 ± 0.11 μmol/g feces, vs. 2.05 ± 0.20 μmol/g feces, *p* < 0.001). cEPO demonstrated an influence on this parameter in the TNBS + cEPO1000 (4.00 ± 0.08 μmol/g feces, *p* < 0.01) but without statistically significant differences in the TNBS + cEPO500 (5.12 ± 0.21 mol/g feces). Likewise, it is not possible to determine a dose-dependent effect since there are no statistically significant differences between both cEPO-treated groups. The fecal hemoglobin was significantly higher in the TNBS + cEPO500 group and in the TNBS + cEPO1000 groups in comparison to the Sham control group (*p* < 0.001 and *p* < 0.05, respectively). The treatment with cEPO demonstrated a beneficial effect in terms of the hemorrhagic focus; however, it was not statistically significant.

### 3.4. Alkaline Phosphatase

ALP was identified as an indicator of intestinal homeostasis, and its levels were measured and compared among all experimental groups (Figure 2). The TNBS group exhibited the highest ALP concentration at 42.75 ± 1.93 U/L. Treatment with cEPO led to a significant reduction in ALP levels, both at the lower dose (*p* < 0.001 compared to the TNBS group) and the higher dose (*p* < 0.5 compared to the TNBS group). However, the differences between the two cEPO doses were not statistically significant, with the TNBS + cEPO500 group showing a concentration of 31.86 ± 1.45 U/L and the TNBS+ cEPO1000 group at 35.33 ± 1.29 U/L. In contrast, the ALP concentrations in the cEPO1000 and Sham groups were very similar, measuring 26.00 ± 1.00 U/L and 25.50 ± 1.18 U/L, respectively (*p* < 0.001 compared to the TNBS group). The Ethanol group had an ALP concentration of 31.25 ± 1.18 U/L (*p* < 0.001 compared to the TNBS group).

### 3.5. Measurement of Cytokines

The chronic TNBS-induced colitis model is associated with the significant production of pro-inflammatory cytokine TNF-α. This cytokine was measured and compared between all the experimental groups (Figure 3). The TNBS group presented the highest concentration of this pro-inflammatory cytokine (74.13 ± 2.31 pg/mL). The treatment with cEPO at both doses, 500 IU/kg and 1000 IU/kg, allowed the TNF-α levels to significantly decrease compared to the TNBS group (52.26 ± 3.82 pg/mL and 46.13 ± 2.16 pg/mL, *p* < 0.001). Moreover, we can observe a dose-dependent effect of cEPO since the differences between both doses were statistically significant (*p* < 0.01). The control groups, cEPO1000, Ethanol, and Sham, also demonstrated a significant decrease in this cytokine compared to the TNBS group (30.60 ± 0.70 pg/mL, 38.17 ± 0.85 pg/mL, and 39.62 ± 1.72 pg/mL, respectively; *p* < 0.001).

IL-10 is an anti-inflammatory cytokine that, in pathological conditions, can also become dysregulated. IL-10 was measured and compared between all the experimental groups (Figure 4). The TNBS group presented the highest concentration of this anti-inflammatory cytokine (70.31 ± 2.43 pg/mL). It was noticed a decrease in the concentration of IL-10 in both treatment groups, namely TNBS + cEPO 500 and TNBS + cEPO1000 groups (28.80 ± 1.19 pg/mL vs. 52.27 ± 3.86 pg/mL, *p* < 0.001), but more significant in the lower dose. Additionally, we can observe a dose-dependent effect of cEPO since there are statistically significant differences between both treated groups (*p* < 0.001). The control groups, cEPO1000, Ethanol, and Sham, also demonstrated a significant decrease in this cytokine compared to the TNBS group (24.10 ± 0.87 pg/m vs. 42.92 ± 2.57 pg/mL vs. 32.04 ± 0.41 pg/mL, *p* < 0.001, compared to TNBS group).

### 3.6. Renal and Hepatic Function

To evaluate renal function, the concentrations of urea and creatinine were measured and compared between all the experimental groups. (Figure 5 and Figure 6). Relatively to urea, It was only possible to identify a significant reduction in the cEPO1000 (66.50 ± 2.50 mg/dL) group when compared to the TNBS group (*p* < 0.05). However, the TNBS + cEPO1000 group presented the highest value, with 84.40 ± 0.60 mg/dL. The cEPO1000 group showed the lowest value (66.50 ± 2.50 mg/dL).

In terms of creatinine, the TNBS group showed the highest concentration of this marker (0.50 ± 0.03 mg/dL). A significant decrease in the concentration of creatinine was observed in both treated groups when compared to the TNBS group. The TNBS + cEPO500 group with 0.40 ± 0.01 mg/dL (*p* < 0.01), whereas the TNBS + cEPO1000 group revealed 0.41 ± 0.02 mg/dL (*p* < 0.05). The differences between doses of cEPO were not statistically significant. The cEPO1000 and Sham control groups showed a statistically significant decrease in creatinine concentration compared to the TNBS group (0.33 ± 0.01 mg/dL and 0.34 ± 0.01 mg/dL, *p* < 0.001, respectively). However, the Ethanol group did not exhibit a significant decrease in creatinine levels (0.42 ± 0.00 mg/dL).

The hepatic function was evaluated by measuring the serum ALT concentration across all experimental groups (Figure 7). In the TNBS group, the ALT level was the highest, at 35.25 ± 2.02 U/L. Treatment with cEPO, at both doses of 500 UIIU/kg/day and 1000 UIIU/kg/day, led to a significant reduction in ALT levels compared to the TNBS group (24.43 ± 1.76 U/L vs. 22.83 ± 1.60 U/L, *p* < 0.001, respectively). However, the differences in ALT levels between the two cEPO doses were not statistically significant. The control groups, Ethanol and Sham, exhibited a substantial decrease in ALT levels compared to the TNBS group (24.33 ± 0.67 U/L vs. 20.80 ± 0.66 U/L; *p* < 0.01 and *p* < 0.001, respectively). Notably, the cEPO1000 group did not show a significant decrease in ALT levels (30.00 ± 0.58 U/L).

### 3.7. Macroscopical Evaluation

The macroscopical evaluation of the colons (Figure 8) revealed that maximal damage was observed in the TNBS group (1.50 ± 0.29), corresponding to localized hyperemia without ulcers. The TNBS + cEPO500 group also presented localized hyperemia, However, with less severity (0.88 ± 0.30) than the TNBS group. The TNBS + cEPO1000 and the Ethanol group presented only slow hyperemia with (0.50 ± 0.38 and 0.25 ± 0.25, respectively). The cEPO1000 and Sham control groups did not present any damage to the colon tissue. We cannot observe a dose-dependent effect of cEPO since there are no statistically significant differences between both treated groups.

### 3.8. Microscopical Assessment of Colitis Severity

Histopathological evaluation of the colon can be represented by the illustrative images presented in Figure 9. Based on inflammatory cell infiltration and tissue damage, the histopathological analysis enables the evaluation of colonic injury. The colons of the TNBS group presented a considerable infiltration of inflammatory cells. These cells were present mostly in the mucosa and submucosa layers, foci of ulceration with necrosis, and tissue disruption. Both cEPO-treated groups present a decreased level of inflammatory cell infiltration along with the area of epithelial ulceration and tissue disruption. Still, cEPO showed a significant anti-inflammatory effect when used at the highest dose (TNBS + cEPO1000), having the least rate of cell infiltration.

For a more objective assessment of colitis severity through microscopic examination, a final histopathological score was generated (Figure 10). As expected, the TNBS group displayed the highest histopathological score, measuring 10.00 ± 0.97. In contrast, the cEPO1000 (0.0 ± 0.0), ethanol (0.50 ± 0.50), and sham (0.40 ± 0.40) groups exhibited a substantial reduction in comparison to the TNBS group (*p* < 0.001). The cEPO-treated groups, TNBS + cEPO500 (7.67 ± 0.50) and TNBS + cEPO1000 (6.00 ± 1.17), displayed a decrease in the histopathological score compared to the TNBS group (*p* < 0.05 and *p* < 0.001, respectively). Furthermore, a dose-dependent effect of cEPO is evident as there are statistically significant differences between both treated groups (*p* < 0.001).

### 3.9. Safety Assessment of Colitis Severity

#### Cardiovascular Function

For the assessment of the adverse effects, the red blood cells (RBCs) and the hematocrit in percentage were measured (Figure 11 and Figure 12). There were no statistically significant differences, considering a *p* < 0.05. Consequently, it is not possible to determine a significant effect of cEPO on the RBCs. The groups treated with cEPO, namely TNBS + cEPO500 and TNBS + cEPO1000, had similar values compared to the Sham group (6.52 ± 0.19 µL vs. 6.74 ± 0.13 µL vs. 6.28 ± 0.16 µL, respectively).

In terms of hematocrit (Figure 12), the highest value was noticed in the TNBS + cEPO1000 group (49.95 ± 0.16%). The groups TNBS + cEPO500, cEPO,1000, and Sham present similar values (48.53 ± 1.13% vs. 49.65 ± 0.14% vs. 49.40 ± 0.29%), without statistically significant differences between all groups, considering a *p* < 0.05. Therefore, it is not possible to determine a significant effect of cEPO on the hematocrit.

## 4. Discussion

In recent years, the cEPO molecule has garnered significant attention for its potential to provide protective and anti-inflammatory effects across various diseases [25]. Derived from the modification of the EPO molecule, cEPO is a non-erythropoietic erythropoietin carbamoyl derivative [25]. EPO plays a crucial role in regulating the production, survival, and differentiation of red blood cells into mature erythrocytes within the human and murine bone marrow by binding to EPOR [26,27,28,29,30]. However, EPO is associated with high side effects due to its stimulation of erythropoiesis, which poses a risk of severe cardiovascular events [31,32]. To mitigate these potential risks and side effects, a promising approach is to utilize the cEPO molecule [33].

Our research group tested EPO in IBD and has revealed a positive anti-inflammatory effect, with no hematocrit alterations. Since it was evaluated in an acute model of colitis, and the experiment took four days in total, it would be interesting to evaluate the EPO derivate at the same doses but now in a chronic animal model of IBD, which is the objective of this experiment.

The rate of carbamoylation is influenced by the number of lysine residues attached to the molecule [34]. The primary mechanism of carbamoylation involves the covalent binding of isocyanic acid to the N-terminal free lysine residues [34]. Upon tryptic hydrolysis of rHuEPO, nineteen peptides and two amino acids are produced due to the presence of twelve arginine and eight lysine residues in this glycoprotein. The purification rate is determined based on the availability of free lysine residues, as indicated by their reaction with TNBS []. TNBS serves as a rapid and sensitive assay reagent for determining free amino groups [].

Regarding disease manifestation, the control group exhibited the most severe prognosis, as evidenced by clinical signs, such as altered stool consistency and anus appearance, marked by diarrhea and/or softer stools, along with severe edema around the anus. These observations confirm the successful induction of experimental colitis with TNBS [35,36]. However, treatment with cEPO demonstrated an improvement in these signs. The determination of fecal hemoglobin is useful in the detection of colorectal diseases, as well as other possible lesions that are accompanied by bleeding 22The treatment with cEPO had a positive influence on decreasing the levels of fecal hemoglobin in comparison to the disease control group. Although there was a reduction of this biomarker, compared to the TNBS group, only the highest dose had statistical significance (*p* < 0.01). These results are in accordance with the results of EPO in the acute model. The treatment with cEPO was capable of reducing the concentration of fecal hemoglobin [27]. ALP has been studied in inflammatory disorders such as IBD. This is a biomarker that plays a vital role in maintaining gut homeostasis [37]. The treatment with cEPO demonstrated a beneficial effect compared to the TNBS group. Actually, the group treated with the lowest dose of cEPO had lower ALP values (*p* < 0.001, compared to the TNBS group). On the other hand, a higher ALP expression is associated with increased intestinal inflammation. This fact was corroborated by the TNBS group, which presented the highest ALP values. According to the results, an increased concentration of ALP may indicate an inflammatory response in the TNBS group that was reduced with cEPO treatment.

As a pro-inflammatory cytokine, it is expected that TNF-⍺ was in higher concentration in the disease group. This fact was confirmed since the TNBS group presented the highest concentration of this biomarker. In contrast, groups treated with cEPO (TNBS + cEPO500 and TNBS + cEPO1000) had lower concentrations (*p* < 0.001). The administration of cEPO showed a beneficial effect on this parameter, taking into account the significant reduction in TNF-⍺ levels in both doses used (*p* < 0.001). Additionally, to sustain this affirmation, the cEPO1000, Ethanol, and Sham control groups presented significantly lower levels of this marker in comparison to the TNBS group (*p* < 0.001). Among the two doses tested, there were observed statistically significant differences, demonstrating a dose-dependent effect (*p* < 0.01). Furthermore, in the acute model of colitis, TNF-⍺ concentrations were higher in the TNBS group compared to the EPO-treated groups, proving its pro-inflammatory effect [27]. Concerning IL-10, the highest concentration of IL-10 was observed in the TNBS group. This fact may be related to the adjustment of the immune system to chronic situations. Indeed, in long-term inflammations, the organism begins to produce anti-inflammatory cytokines to control the inflammatory process [38]. Treatment with cEPO at 500 IU/kg and 1000 IU/kg had a positive influence on significantly decreasing the levels of this anti-inflammatory cytokine in both doses (*p* < 0.001), with a dose-dependent effect (*p* < 0.001). In general, there were no significant differences between the cEPO-treated groups and the control groups. According to the results obtained, cEPO has shown a beneficial effect on the inflammatory response, which is consistent with preclinical studies developed with EPO in the acute mode [26,27,28,29,30].

As a marker of hepatic function, the fact that lower values were obtained in the TNBS + cEPO500 and TNBS + cEPO1000 IU/kg groups (*p* < 0.001) is a favorable indicator to exclude hepatic damages [39,40]. The treatment groups showing ALT levels close to those of the Sham group may serve as a positive indicator of liver function [41]. Since IBD can lead to various extra-intestinal manifestations, it is crucial to emphasize the periodic evaluation of renal functions as well. Observing the urea concentration, there were only significant differences between the cEPO1000 group and the TNBS group (*p* < 0.05). Inversely to what was expected, the values of urea reached in the TNBS group were similar to those in the Ethanol and Sham control groups. This may be explained by the lower concentration of TNBS used compared to the acute model (1% vs. 2.5%). In fact, the highest value obtained was in TNBS + cEPO1000, which was also not expected, but it is in agreement with the results of EPO in the acute model [27]. Although there are high urea values in groups treated with cEPO, it did not demonstrate statistical significance. However, it may indicate a possible negative influence of cEPO in this biomarker. Regarding creatinine, it was possible to observe that the disease control group demonstrated the highest concentration of this marker. The treatment with cEPO at 500 and 1000 IU/kg significantly reduced the concentration of creatinine (*p* < 0.01 and *p* < 0.05, respectively), which is in accordance with the results of our research group in the acute colitis model with EPO [27]. However, the values obtained in the TNBS + cEPO500 and TNBS + cEPO1000 groups were higher than those observed in the cEPO1000 and Sham control groups, nevertheless, without statistical significance. It is possible to conclude that cEPO does not promote renal and/or hepatic changes because the cEPO1000 group had no elevated levels of these biochemical markers.

Based on the scoring of gross morphologic damage described by Morris et al. (1989) [21], several macroscopic parameters can be analyzed in the colon of the *mice*. Only cEPO1000 and Sham control groups presented no macroscopical alterations. The treated groups had a lower score compared to the TNBS group, which presented the peak of colon damage, corresponding to localized hyperemia, but no ulcers, however, without statistical significance. According to the results obtained, it is not possible to determine a strong beneficial effect of cEPO. Actually, the colon of treated groups showed focal hyperemia similar to the TNBS control group. However, the colons of the TNBS group revealed a slight thickening, as described by Anthoniou et al., that was not present in the treated groups. These results may demonstrate a slight positive effect of cEPO.

Through the microscopical evaluation, it was observed a beneficial effect by cEPO in the treated groups with cEPO at 500 and 1000 IU/kg (*p* < 0.05 and *p* < 0.001, respectively). A reduction in the histopathological score was observed, with a dose-dependent effect (*p* < 0.001). Data available in the literature regarding histopathological evaluation show that EPO significantly attenuates the acute inflammatory response [42]. Nevertheless, in a chronic context, it was not possible to confirm it, as shown in this experiment [27].

Hematocrit represents the proportion of red blood cells (RBCs) or erythrocytes in the total blood volume [43]. As it is well-established, EPO promotes erythropoiesis, resulting in an augmented production of RBCs and subsequently leading to an elevation in the hematocrit level [27].

The hematopoietic effect associated with EPO can be inhibited through its carbamylation [43,44]. Observing the results obtained, there was no change in this biomarker between the groups treated with cEPO and the cEPO1000 and Sham control groups. Additionally, according to reference levels of hematocrit in *CD-1 mice* (43.9 and 53.3%), all groups showed normal hematocrit levels [45]. These data suggest that cEPO does not increase the risk of adverse events related to EPO molecules [27]. Therefore, it suggests that results suggest that treatment with cEPO is safer and involves fewer risks and adverse reactions like severe cardiovascular events.

The gut microbiome is intimately linked with colitis, playing a significant role in the development and progression of the disease. Future treatments for colitis are likely to involve strategies aimed at restoring a healthy gut microbiome to reduce inflammation and improve patient outcomes. Inflammation in the gut can alter the environment, making it more favorable for harmful bacteria and less hospitable to beneficial ones. The gut microbiome plays a crucial role in regulating the immune system. Dysbiosis can lead to an inappropriate immune response, where the immune system may overreact to harmless substances, leading to chronic inflammation characteristic of colitis. The gut microbiome also contributes to the maintenance of the gut barrier. When this barrier is compromised, as is often the case in colitis, harmful bacteria and their products can penetrate the gut lining, triggering inflammation. Current treatments for colitis can be effective at managing symptoms and inducing remission; they do not target the microbiome directly. There is growing interest in developing treatments that modulate the gut microbiome to improve outcomes for colitis patients. Some predictions regarding the interaction of treatments with the microbiome include probiotics and prebiotics, fecal microbiota transplantation, microbiome-based therapies, and dietary interventions.

## 5. Conclusions

The anti-inflammatory effects of cEPO were evident in an animal model of colitis induced by TNBS. Based on the outcomes of this study, cEPO successfully restored normal clinical indicators, such as stool consistency and anal appearance, and alleviated anal edema associated with chronic experimental colitis. Furthermore, it positively influenced the expression of biomarkers like fecal hemoglobin, ALT, and ALP. Notably, cEPO significantly reduced the levels of the pro-inflammatory cytokine TNF-α and increased the levels of the anti-inflammatory cytokine IL-10. Importantly, a dose-dependent impact of cEPO on TNF-α expression was observed.

Moreover, cEPO exhibited positive effects on extra-intestinal manifestations linked to IBD without causing any significant adverse effects. It is noteworthy that cEPO plays a crucial role in IBD by mimicking the three-dimensional structure of EPO while inhibiting erythropoiesis, thus reducing EPO-related side effects. This makes cEPO therapeutically advantageous when compared to the conventional molecule, as it maintains the structural resemblance but lacks the erythropoietic function.

These findings strongly suggest that cEPO effectively suppresses chronic inflammation in this experimental colitis model. Therefore, based on the results of this experimental study, it can be concluded that cEPO presents a highly promising pharmacological approach for the future treatment of inflammatory-related conditions, including IBD.

## Figures and Tables

**Figure 1 biomedicines-11-02497-f001:**
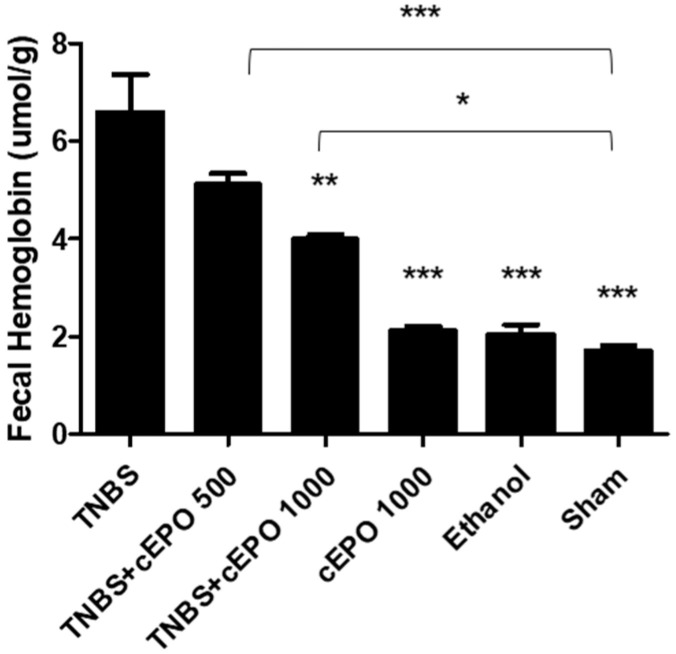
Effect of carbamylated erythropoietin on fecal hemoglobin. Legend: One-way ANOVA and Tukey’s post hoc test; * *p* < 0.05; ** *p* < 0.01; *** *p* < 0.001 compared with TNBS group or between groups.

**Figure 2 biomedicines-11-02497-f002:**
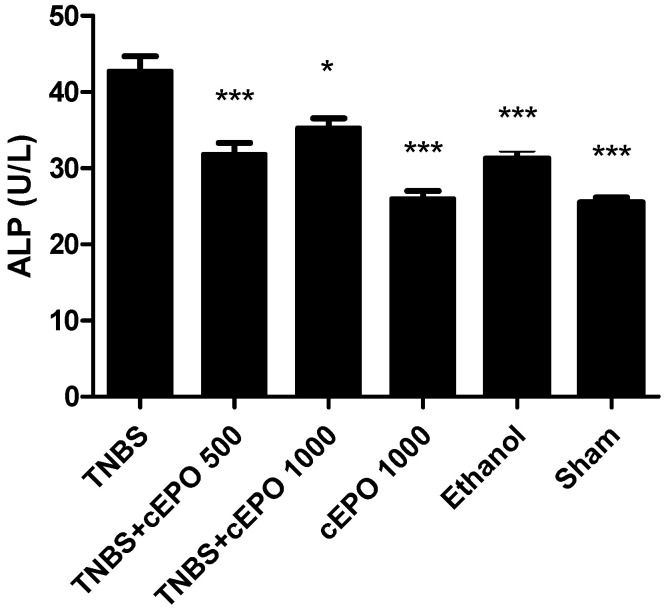
Effect of carbamylated erythropoietin on serum alkaline phosphatase concentration. Legend: One-way ANOVA and Tukey’s post hoc test; * *p* < 0.05; *** *p* < 0.001 compared with TNBS group.

**Figure 3 biomedicines-11-02497-f003:**
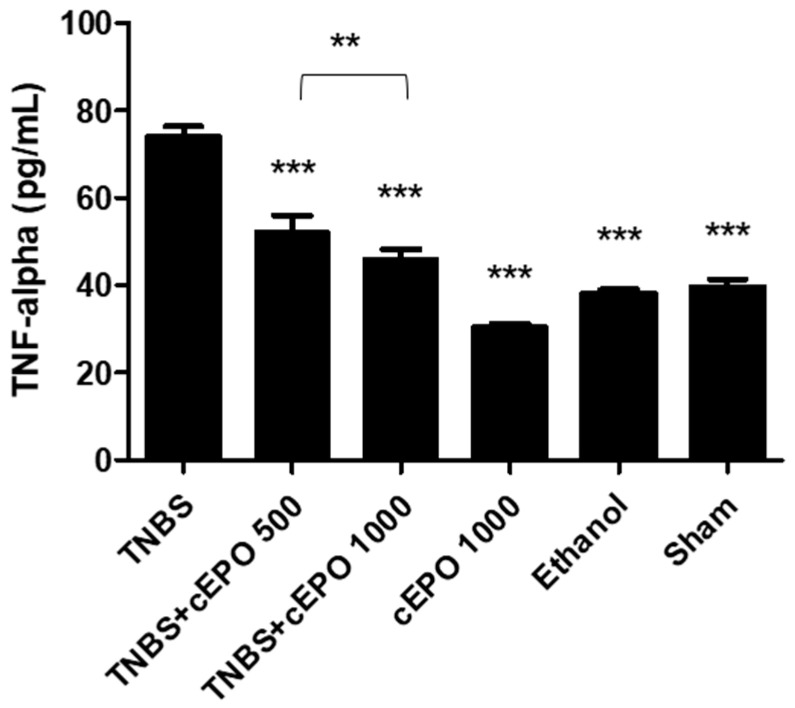
Effect of carbamylated erythropoietin on tumor necrosis factor-alpha concentration. Legend One-way ANOVA and Tukey’s post hoc test; ** *p* < 0.01; *** *p* < 0.001 compared with TNBS group or between groups.

**Figure 4 biomedicines-11-02497-f004:**
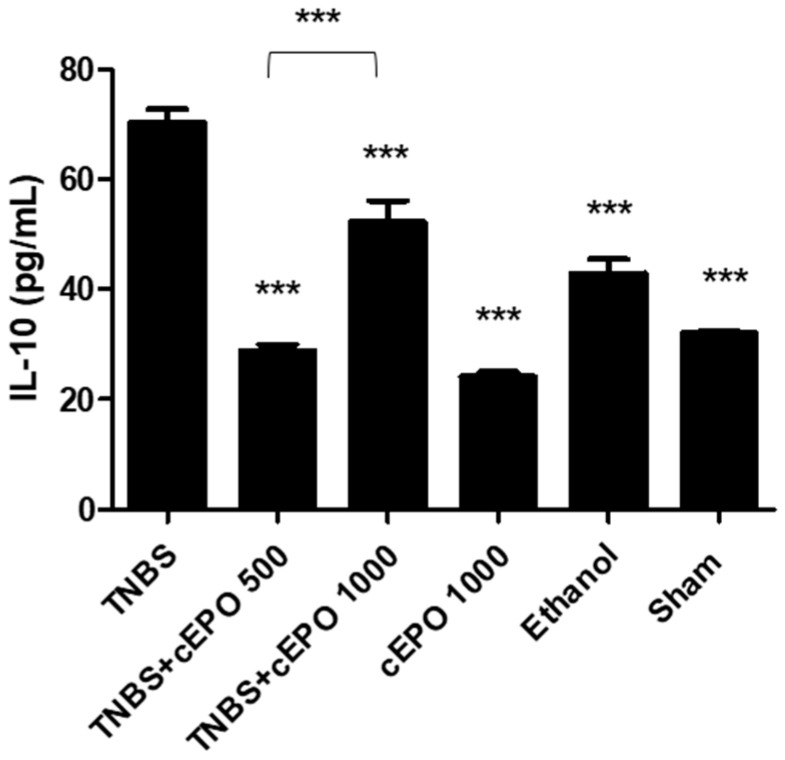
Effect of carbamylated erythropoietin on interleukin-10 concentration. Legend: One-way ANOVA and Tukey’s post hoc test; *** *p* < 0.001 compared with TNBS group or between groups.

**Figure 5 biomedicines-11-02497-f005:**
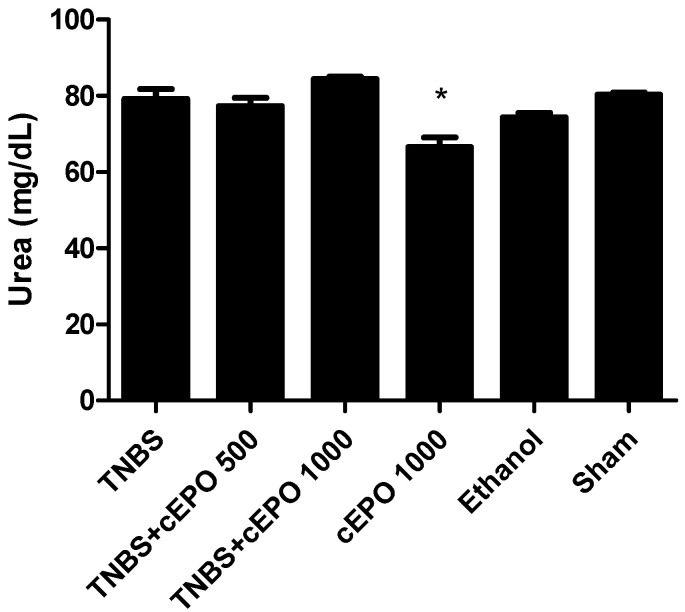
Effect of carbamylated erythropoietin on urea concentration. Legend: One-way ANOVA and Tukey’s post hoc test; * *p* < 0.05 compared with TNBS group.

**Figure 6 biomedicines-11-02497-f006:**
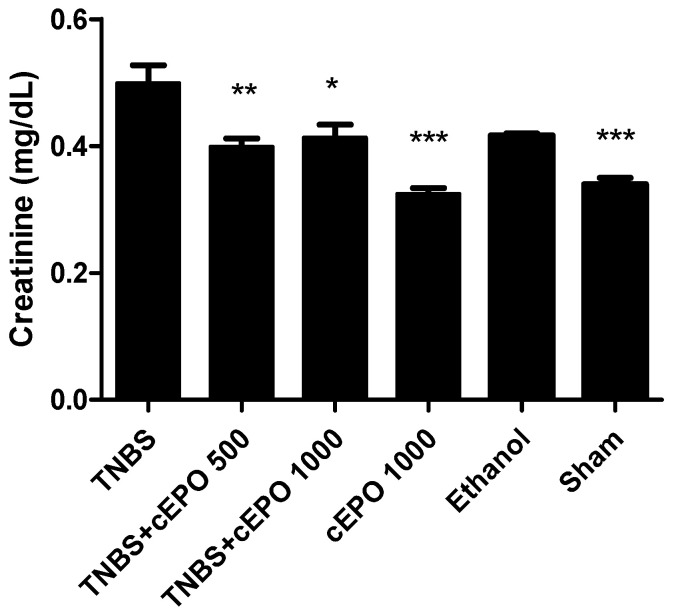
Effect of carbamylated erythropoietin on creatinine concentration. Legend: One-way ANOVA and Tukey’s post hoc test; * *p* < 0.05; ** *p* < 0.01; *** *p* < 0.001 compared with TNBS group.

**Figure 7 biomedicines-11-02497-f007:**
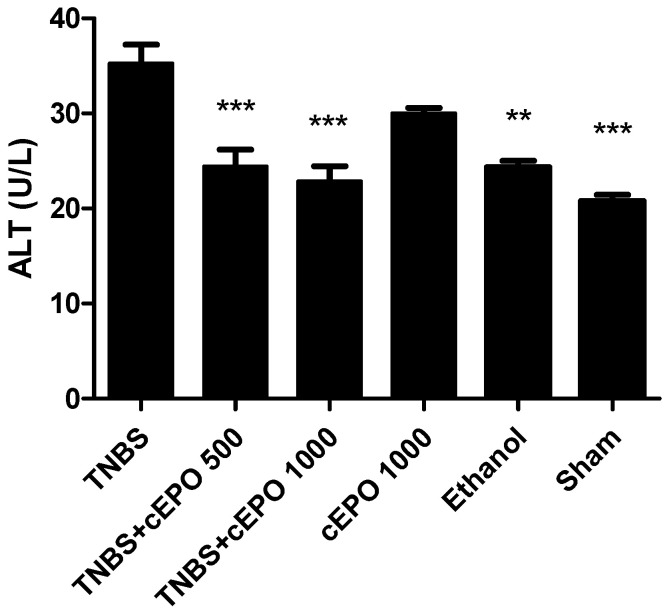
Effect of carbamylated erythropoietin on alanine aminotransferase concentration. Legend: One-way ANOVA and Tukey’s post hoc test; ** *p* < 0.01; *** *p* < 0.001 compared with TNBS group.

**Figure 8 biomedicines-11-02497-f008:**
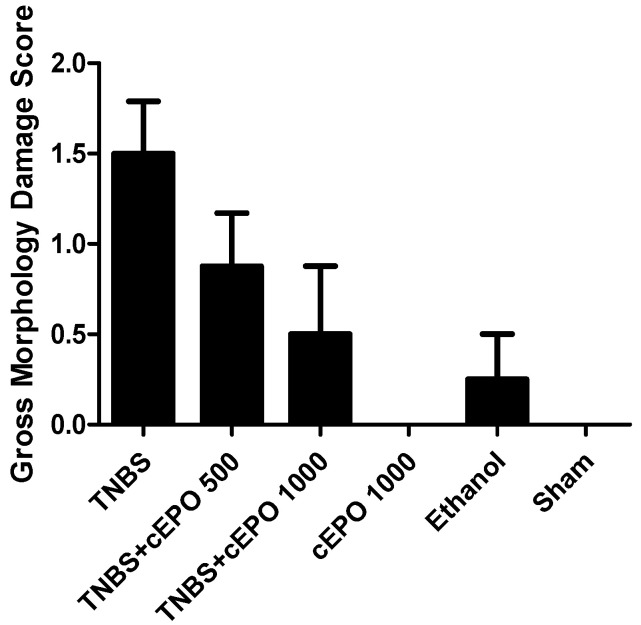
Effect of carbamylated erythropoietin on Gross morphology damage score.

**Figure 9 biomedicines-11-02497-f009:**
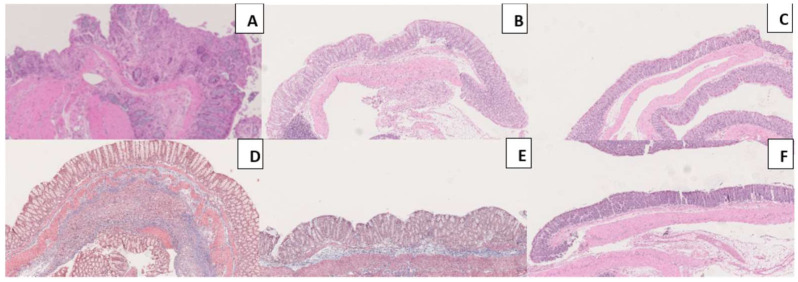
Histopathological analyses of Masson’s trichrome staining 10×. Legend: (**A**) TNBS group; (**B**) ethanol group; (**C**) sham group; (**D**) TNBS + cEPO500 group; (**E**) TNBS + cEPO1000 group; (**F**) cEPO1000 group.

**Figure 10 biomedicines-11-02497-f010:**
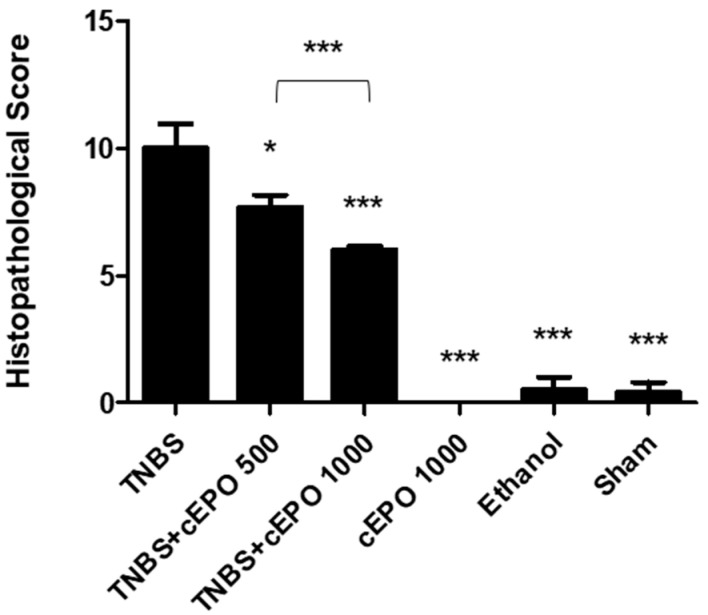
Effect of carbamylated erythropoietin on histopathological score. Legend: One-way ANOVA and Tuckey’s post hoc test. * *p* < 0.05; *** *p* < 0.001; compared with the TNBS group and between the other groups.

**Figure 11 biomedicines-11-02497-f011:**
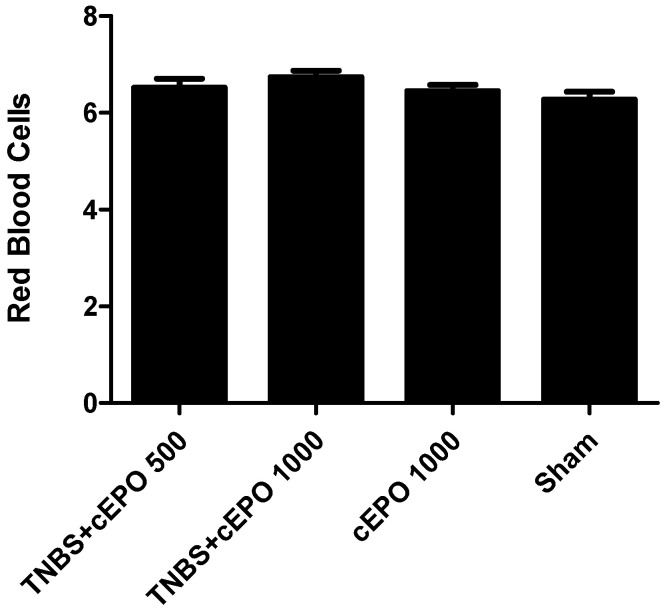
Effect of carbamylated erythropoietin on red blood cell count.

**Figure 12 biomedicines-11-02497-f012:**
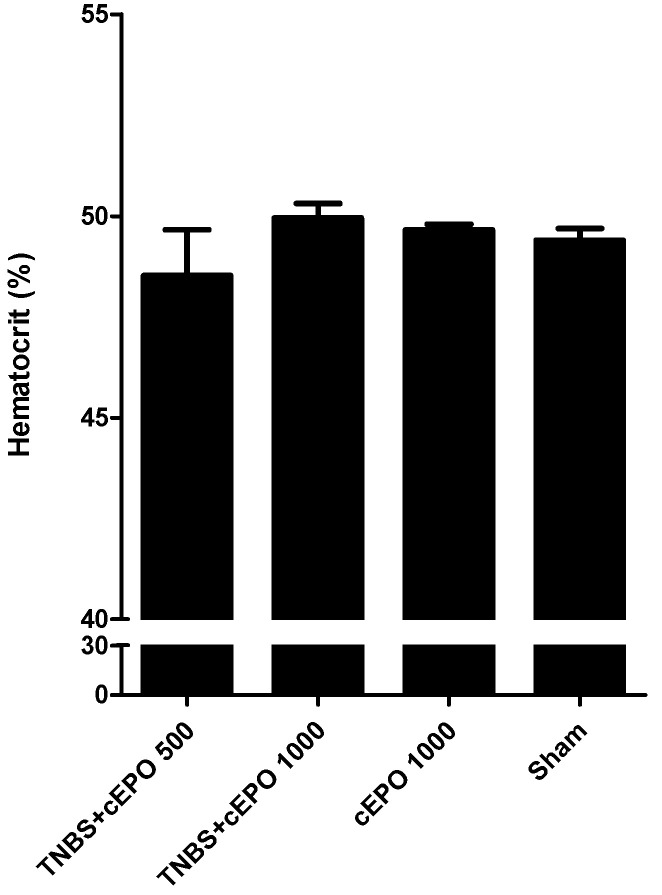
Effect of carbamylated erythropoietin on hematocrit.

## Data Availability

Not applicable.

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
