# Peer review of "Effect of Carbamylated Erythropoietin in a Chronic Model of TNBS-Induced Colitis"

_biomedicines, 2023, doi:10.3390/biomedicines11092497_

Round 1
Reviewer 1 Report
In the present experimental study Silva et al showed that carbamylated erythropoietin (cEPO) has an anti-inflammatory effect on a chemical murine model of ulcerative colitis. Main comments:
1) A linguistic revision is necessary (see for instance doesn’t - - -> does not).
2) Why only female mice were used?
3) Intracolonic TNBS administration: by enema? By colonoscopy?
4) Authors did not describe nor report a reference about the histopathological score that was used. The same applies to the Gross morphology damage score, reported in figure 8.
5) In is unclear whether cEPO was given iv or via enema as TNBS.
6) My most relevant criticism is about side effects. Authors did not evaluate whether cEPO may have a stimulatory effect on epithelial cell growth (as they did not investigate cell proliferation and apoptosis). UC indeed has an increased risk of colon cancer, therefore cEPO must be safe on this side.
see above
Author Response
In the present experimental study Silva et al showed that carbamylated erythropoietin (cEPO) has an anti-inflammatory effect on a chemical murine model of ulcerative colitis. Main comments:
Response: Thank you for your feedback and all the time you spent working on our manuscript. We will answer all the points of criticism described by the reviewer in the same order:
1) A linguistic revision is necessary (see for instance doesn’t - - -> does not).
Response:
The English was already revised.
2) Why only female mice were used?
Response:
According to the literature, there is no trend towards gender. In a systematic review developed by our research group (Silva et al, 2019), the number of articles referring to the use of males or females is similar. Some articles do not even refer to gender. Several authors consider that IBD does not have an associated hormonal component, so it makes sense that there are no differences between genders. However, Scheiffele & Fuss (2001) state that although both males and females can develop TNBS-induced colitis, with the same clinical characteristics, males can develop more expressive and chronic disease, increasing mortality rates. Additionally, males reveal a more territorial instinct, increasing the stress of living together, which could influence the final results. Consequently, we decided to use females.
3) Intracolonic TNBS administration: by enema? By colonoscopy?
Response:
The intracolonic TNBS administration was made by enema. The induction of chronic colitis was developed through weekly intrarectal administrations of TNBS, by enema. The animals were left unfed 24 hours before each administration to facilitate the intrarectal administrations due to the absence of feces. Prior to each intrarectal administration, mice were anesthetized, through an intraperitoneal injection of 40 µL of a solution containing ketamine 100 mg/kg + xylazine 10 mg/ kg. Afterward, 100 µL of TNBS 1 % was administrated intrarectally, with the help of a cannula, carefully inserted until 4 cm in the colon. Then, in order to avoid rectal reflux, the animals were kept in the Trendelenburg position for 1 minute.
4) Authors did not describe nor report a reference about the histopathological score that was used. The same applies to the Gross morphology damage score, reported in figure 8.
Response:
It was a lapse. We agree with the reviewer. Thanks for the detection. We already improved this point as you can verify in the manuscript.
We made reference to the macroscopic score according to Morris et al (1989), but the reference to the histological score (Seamons & colleagues, 2013) was missing.
In order to evaluate colitis severity, macroscopical and microscopical evaluation of colon tissue were performed. For the macroscopical evaluation, the analysis took in account the length of the colon tissue, the presence of hyperemia, adhesions, and/or ulcers, according to Morris et al., 1989. For the microscopical evaluation, a portion of four centimeters of distal colon tissue was fixed in 10 % phosphate-buffered formalin, sectioned at 5 µm, and then stained with hematoxylin and eosin. To increase the possibility of detecting fibrosis, Masson’s trichrome staining was used. Sections of the distal colon were evaluated based on the adapted criteria of Seamons and colleagues (2013). The histopathological score of lesions was partially scored (0–4 increasing in severity) with some parameters, namely: (1) the presence of tissue loss/necrosis; (2) the severity of the mucosal epithelial lesion; (3) inflammation; (4) extent 1—the percentage of the intestine affected in any manner; and (5) extent 2—the percentage of intestine affected by the most severe lesion. The colitis severity was calculated by summing the individual lesions and the extent scores, promoting a final colitis score (max score = 20)
5) In is unclear whether cEPO was given iv or via enema as TNBS.
Response:
The cEPO was administered through intraperitoneal (IP) injections, between days 22–35 of the experiment.
6) My most relevant criticism is about side effects. Authors did not evaluate whether cEPO may have a stimulatory effect on epithelial cell growth (as they did not investigate cell proliferation and apoptosis). UC indeed has an increased risk of colon cancer, therefore cEPO must be safe on this side.
Response:
Our research group has already tested EPO in the acute model (Mateus et al, 2017) and in the chronic model of colitis (Silva et al, 2022) with good results. Taking into account the cEPO profile, we considered that it is important to test this derivative in the first phase, taking into account all the biomarkers already evaluated for EPO, in order to have a comparison. The next step, taking into account the excellent results obtained, will then be to evaluate whether cEPO may have a stimulatory effect on epithelial cell growth, studying cell proliferation and apoptosis. However, IBD is also associated with extraintestinal manifestations, resulting from inflammation, and in this sense, we evaluated liver and kidney biomarkers to test the safety of the molecule. Based on the results, cEPO can demonstrate an anti-inflammatory effect, without compromising liver and kidney safety.
Reviewer 2 Report
The manuscript submitted to Biomedicines by Silva et al., is an interesting in vivo study aiming to investigate the effects of carbamoyled erythropoietin on a model of TNBS-induced colitis. The reviewer would like to offer the following points for consideration by the authors.
1. How was the number of animals determined? (power calculation?).
2. How was the dosing determined (rationale/reference).
3. What was the rationale for the time-points selected for data collection/ measurements?
4. Did the authors consider the feed intake in their analyses?
5. It would be interesting to discuss the microbiome aspect and how it interplays with the concept of colitis. What would be the plausible prediction as per the interaction of the treatment with the microbiome?
6. In terms of the immune response diet may extend an effect. It would be interesting to include a brief discussion in this regard.
7. It would be helpful to include the hypothesis of the study in the end of the introduction section.
The English language could improve in terms of proper grammar/syntax, typos and flow. Having the manuscript proofread by a native English speaker would help.
Author Response
The manuscript submitted to Biomedicines by Silva et al., is an interesting in vivo study aiming to investigate the effects of carbamoyled erythropoietin on a model of TNBS-induced colitis. The reviewer would like to offer the following points for consideration by the authors.
Response: Thank you for your feedback and all the time you spent working on our manuscript. We will answer all the points of criticism described by the reviewer in the same order:
- How was the number of animals determined? (power calculation?).
Response: Our N is defined according to the 3Rs policy, in order to reduce, refine, and replace and also in order to our results to have statistical significance. Therefore, the N of the control groups is lower than that of the experimental groups.
- How was the dosing determined (rationale/reference).
Response: Our research group has already tested EPO in the acute model (Mateus et al, 2017) and in the chronic model (Silva et al, 2022) of colitis with good results. In this sense, we considered that it is important to test this derivative in the same conditions, i.e., in the same dose and taking into account all the biomarkers already evaluated for EPO, in order to have a comparison.
- What was the rationale for the time-points selected for data collection/ measurements?
Response: Prior to testing molecules in a model of colitis, we developed and validated a chronic TNBS induced model. This TNBS-induced colitis model was monitored for 6 weeks. A scheme of multiple TNBS administrations was performed since the aim was to achieve a chronic pattern of induced colitis and to identify the week in which the damage becomes chronic. Clinical manifestations of chronic colitis usually peak within 2 weeks and may be followed by partial recovery or death. These results were expected and are compatible with the correct induction of colitis. Our findings allow us to conclude that TNBS-induced chronic colitis should be developed in 4 weeks, providing a chronic intestinal inflammation model. Indeed, the parameters under evaluation, such as clinical manifestations; biochemical markers, including fecal hemoglobin and pro- and anti-inflammatory cytokine levels; and macroscopic evaluation, corroborate that the chronic illness pattern is observed from week 4 after the induction. Some mice died during the early days of the study, possibly because they did not resist the aggravation of the disease in its acute phase. On the other hand, the remaining mice resisted and progressed to the chronic phase of the disease, showing the same symptoms but more lightly. In this sense, we induced the disease for 4 weeks, and in the last 2 weeks, we tested the molecules.
- Did the authors consider the feed intake in their analyses?
Response:
It was a lapse. We agree with the reviewer. Thanks for the detection. We already improved this point as you can verify in the manuscript.
We didn't consider it, but thanks for the detection. In the next experiment, we will take feed consumption into account. However, this factor had no influence on our results since all animals in all groups were in the same experimental conditions. Additionally, on induction days, food was withdrawn in all groups, even those not submitted to 1% TNBS IR administration, precisely in order to ensure equal conditions among all groups.
- It would be interesting to discuss the microbiome aspect and how it interplays with the concept of colitis. What would be the plausible prediction as per the interaction of the treatment with the microbiome?
- In terms of the immune response diet may extend an effect. It would be interesting to include a brief discussion in this regard.
Response:
The gut microbiome is intimately linked with colitis, playing a significant role in the development and progression of the disease. Future treatments for colitis are likely to involve strategies aimed at restoring a healthy gut microbiome to reduce inflammation and improve patient outcomes. Inflammation in the gut can alter the environment, making it more favorable for harmful bacteria and less hospitable to beneficial ones. The gut microbiome plays a crucial role in regulating the immune system. Dysbiosis can lead to an inappropriate immune response, where the immune system may overreact to harmless substances, leading to chronic inflammation characteristic of colitis. The gut microbiome also contributes to the maintenance of the gut barrier. When this barrier is compromised, as is often the case in colitis, harmful bacteria and their products can penetrate the gut lining, triggering inflammation. Current treatments for colitis can be effective at managing symptoms and inducing remission, they do not target the microbiome directly. There is growing interest in developing treatments that modulate the gut microbiome to improve outcomes for colitis patients. Some predictions regarding the interaction of treatments with the microbiome include: probiotics and prebiotics, fecal microbiota transplantation, microbiome-based therapies and dietary interventions
- It would be helpful to include the hypothesis of the study in the end of the introduction section.
Response:
Thanks for the observation. Alteration included in the manuscript.
Reviewer 3 Report
It is a comprehensive study regarding a promising drug candidate, carbamylated erthropoietin that could be used to treat inflammatory bowel disease. The study was conducted using a chronic mouse model of TNBS-induced colitis, focusing on clinical signs, fecal hemoglobin, biomarkers such as alkaline phosphate levels (as biomarker of intestinal homeostasis), cytokines, renal function, etc. In general, the study design is logical and the presented data are of satisfactory quality. Just have one minor comment: it would be nice if the authors monitored pharmacokinetcis of carbamylated erthropoietin.
see comment above "Minor editing of English language required".
Author Response
It is a comprehensive study regarding a promising drug candidate, carbamylated erthropoietin that could be used to treat inflammatory bowel disease. The study was conducted using a chronic mouse model of TNBS-induced colitis, focusing on clinical signs, fecal hemoglobin, biomarkers such as alkaline phosphate levels (as biomarker of intestinal homeostasis), cytokines, renal function, etc. In general, the study design is logical and the presented data are of satisfactory quality. Just have one minor comment: it would be nice if the authors monitored pharmacokinetcis of carbamylated erthropoietin.
Response: Thank you for your feedback and all the time you spent working in our manuscript. Our research group has already tested EPO in the acute model (Mateus et al, 2017) and in the chronic model (Silva et al, 2022) of colitis with very good results. In this sense, we considered that, in the first phase, it is important to test this derivative in the same conditions and taking into account all the biomarkers already evaluated for EPO, in order to make a comparison. However, our goal is to deepen the investigation, which includes pharmacokinetic studies.
Round 2
Reviewer 1 Report
Answers were fine
Author Response
Thank you for your feedback and all the time you spent working on our manuscript.
We already improved the manuscript as you can verify.

Reviewer 2 Report
The authors have made a reasonable effort in addressing reviewer's points.
English is OK proofreading is recommended for typos and flow.
Author Response
Thank you for your feedback and all the time you spent working on our manuscript.
The English was already revised as you can verify in the manuscript.